# Tankyrase Inhibition Attenuates Cardiac Dilatation and Dysfunction in Ischemic Heart Failure

**DOI:** 10.3390/ijms231710059

**Published:** 2022-09-02

**Authors:** Hong Wang, Heli Segersvärd, Juuso Siren, Sanni Perttunen, Katariina Immonen, Riikka Kosonen, Yu-Chia Chen, Johanna Tolva, Mirjami Laivuori, Mikko I. Mäyränpää, Petri T. Kovanen, Juha Sinisalo, Mika Laine, Ilkka Tikkanen, Päivi Lakkisto

**Affiliations:** 1Minerva Foundation Institute for Medical Research, 00290 Helsinki, Finland; 2Zebrafish Unit, HiLIFE and Department of Anatomy, University of Helsinki, 00014 Helsinki, Finland; 3Transplantation Laboratory, Department of Pathology, University of Helsinki, 00014 Helsinki, Finland; 4Department of Vascular Surgery, Abdominal Center, Helsinki University Hospital and University of Helsinki, 00014 Helsinki, Finland; 5Department of Pathology, University of Helsinki and Helsinki University Hospital, 00014 Helsinki, Finland; 6Atherosclerosis Research Laboratory, Wihuri Research Institute, 00290 Helsinki, Finland; 7Heart and Lung Center, University of Helsinki and Helsinki University Hospital, 00014 Helsinki, Finland; 8Abdominal Center, Nephrology, University of Helsinki and Helsinki University Hospital, 00014 Helsinki, Finland; 9Department of Clinical Chemistry, University of Helsinki and Helsinki University Hospital, 00014 Helsinki, Finland

**Keywords:** ischemic heart failure, myocardial infarction, tankyrase, miRNA, Wnt/β-catenin signaling

## Abstract

Hyperactive poly(ADP-ribose) polymerases (PARP) promote ischemic heart failure (IHF) after myocardial infarction (MI). However, the role of tankyrases (TNKSs), members of the PARP family, in pathogenesis of IHF remains unknown. We investigated the expression and activation of TNKSs in myocardium of IHF patients and MI rats. We explored the cardioprotective effect of TNKS inhibition in an isoproterenol-induced zebrafish HF model. In IHF patients, we observed elevated TNKS2 and DICER and concomitant upregulation of miR-34a-5p and miR-21-5p in non-infarcted myocardium. In a rat MI model, we found augmented TNKS2 and DICER in the border and infarct areas at the early stage of post-MI. We also observed consistently increased TNKS1 in the border and infarct areas and destabilized AXIN in the infarct area from 4 weeks onward, which in turn triggered Wnt/β-catenin signaling. In an isoproterenol-induced HF zebrafish model, inhibition of TNKS activity with XAV939, a TNKSs-specific inhibitor, protected against ventricular dilatation and cardiac dysfunction and abrogated overactivation of Wnt/β-catenin signaling and dysregulation of miR-34a-5p induced by isoproterenol. Our study unravels a potential role of TNKSs in the pathogenesis of IHF by regulating Wnt/β-catenin signaling and possibly modulating miRNAs and highlights the pharmacotherapeutic potential of TNKS inhibition for prevention of IHF.

## 1. Introduction

Myocardial infarction (MI) is a leading cause of death worldwide [1]. Cardiac remodeling after MI results in fibrotic scar formation, cardiomyocyte loss, and cardiac dysfunction, progressively leading to heart failure (HF) in humans [2]. Currently, limited therapies are available due to the minimal regenerative capacity of human hearts. Better understanding of the molecular mechanisms underlying ischemic heart failure (IHF) can provide potential new therapeutic approaches.

MicroRNAs (miRNAs), known posttranscriptional regulators of gene expression, critically control the pathophysiological processes of MI [3]. Modulation of miRNAs after MI can reduce infarct size and attenuate cardiac dysfunction [4,5]. Thus, targeting miRNAs may have therapeutic potential for cardiac repair after MI. The biosynthesis and function of miRNAs are tightly regulated at multiple levels, including the processing of pre-miRNAs into mature miRNAs by DICER and the recruitment of miRNAs to target mRNAs by ARGONAUTE (AGO) proteins [6]. The abundance and activity of DICER are tightly regulated transcriptionally and post-translationally, and defects in such regulation impair miRNA biogenesis and can initiate pathological processes [7]. DICER can be phosphorylated, SUMOylated, and proteolyzed into truncated fragments (∼50 and ∼170 kDa), which affect its nuclear translocation and function [8,9,10]. The stability of AGOs can also be modulated by posttranslational modification, such as poly-ADP-ribosylation (PARylation), which affects the expression and activity of miRNAs [11].

Wnt/β-catenin signaling is crucial during cardiac development but relatively silent in adult hearts in physiological conditions [12,13]. MI, however, activates Wnt/β-catenin signaling in several cardiac cell types, including cardiomyocytes, endothelial cells, epithelial cells, and fibroblasts [14,15,16]. Conflicting studies elucidating the role of Wnt/β-catenin signaling after cardiac injury have been reported, depending on the type of cardiac injury (acute vs. chronic), and the type of cardiac cells investigated [17]. Although the majority of studies have demonstrated the beneficial effects of inhibition of Wnt/β-catenin signaling on infarct size and cardiac function after MI [17], a growing body of studies have shown that Wnt/β-catenin activates cardiac cell proliferation to promote cardiac repair after MI [14,17,18]. Wnt/β-catenin signaling is negatively regulated by the β-catenin destruction complex consisting of the AXIN scaffold protein, the adenomatous polyposis coli (APC) tumor suppressor, and two serine–threonine kinases (glycogen synthase kinase 3 (GSK3) and casein kinase 1α (CK1α)) [19]. The destruction complex targeting β-catenin for proteasomal degradation is constitutively active to keep signaling completely off in the absence of a Wnt ligand. The key component of the β-catenin destruction complex, AXIN, is destabilized by tankyrases (TNKSs), which in turn disrupts the complex and thereby activates Wnt signaling [20].

TNKS1 and TNKS2 are poly (ADP-ribose) polymerases (PARPs), of which poly ADP-ribosylates (PARylates) target proteins for proteasomal degradation and consequently modulate diverse cellular processes [21]. Notably, genome-wide association studies indicate that *TNKS* and *TNKS2* (encoding TNKS1 and TNKS2) are associated with MI, ischemic stroke, and hypertension in humans [22,23], suggesting a crucial role for TNKSs in the pathogenesis of cardiovascular disease (CVD). These observations led us to explore the role of TNKSs in pathophysiological processes of IHF and the therapeutic potential of TNKS inhibition for IHF.

## 2. Results

### 2.1. Cardiac TNKS2 and DICER Are Augmented in IHF Patients

We investigated the cardiac expression of TNKS1 and TNKS2 in IHF patients in comparison with healthy controls. The protein level of TNKS2 was significantly increased (*p* = 0.01) in the left ventricle (LV) of IHF patients (Figure 1A). Unlike TNKS2, the protein level of TNKS1 remained unchanged (Figure 1A). As TNKS1 PARylates AXIN1 for degradation and subsequently stabilizes β-catenin (β-CAT) [20], we further investigated the expression of AXIN1 and the active form of β-CAT. In line with TNKS1, both AXIN1 and active β-catenin (Act-β-CAT) showed no significant difference in protein levels between IHF patient hearts and healthy controls (Figure 1B). TNKSs have been shown to PARylate DICER, which affects the activity of miRNAs [11,21]. We therefore examined the expression of DICER. Consistent with a previous study [24], we identified three isoforms of DICER, including a full-length form (∼250 kD) and two shorter alternative isoforms (∼113 kD and ∼93 kD) (Figure 1C). In addition, we detected a truncated DICER (ΔDICER, ∼50 kDa) (Figure 1C). Along with the elevation of TNKS2, full-length DICER was enriched (*p* = 0.003) in IHF patient hearts (Figure 1C). Concomitant with the accumulation of DICER, ΔDICER was also markedly enriched (*p* = 0.005) in IHF patient hearts (Figure 1C). Taken together, ischemic injury induced the augmentation of TNKS2 and DICER in the LV of IHF patients. 

### 2.2. Cardiac miR-34a-5p and miR-21-5p Are Upregulated in IHF Patients

Enrichment of full-length DICER in IHF patient hearts prompted us to explore the expression of miRNAs. We quantitatively evaluated the expression levels of selected miRNAs associated with MI and LV remodeling, miR-34a-5p, miR-21-5p, miR-208a-3p, miR-181a-5p, miR-133a-3p, and miR-19b-3p. Of these, miR-34a-5p (*p* = 0.045) and miR-21-5p (*p* = 0.034) were markedly upregulated in the LV of IHF patients in comparison with the controls (Figure 1D). However, the expression levels of miR-208a-3p, miR-181a-5p, miR-133a-3p, and miR-19b-3p exhibited no significant difference between patient hearts and healthy controls (Figure 1D). The data indicate that the expression of miR-34a-5p and miR-21-5p is dysregulated in IHF patient hearts, suggestive of a potential role in LV remodeling.

### 2.3. MI Stimulates the Expression and Activation of TNKSs in Rat Hearts

We next investigated the cardiac expression and activity of TNKSs in MI rats compared to sham-operated controls. MI elevated the protein levels of TNKS1 in the border area at 1 week (border vs. sham, *p* < 0.001; border vs. remote, *p* = 0.003) (Figure 2A) and 4 weeks (border vs. sham, *p* = 0.01; border vs. remote, *p* = 0.02) (Figure 2B) post-MI when infarcted hearts underwent cardiac remodeling and scar formation. Furthermore, TNKS2 showed a tendency toward increase at 1 week (border vs. sham, *p* = 0.08; border vs. remote, *p* = 0.05) (Figure 2A) and a significant increase at 4 weeks post-MI (border vs. sham, *p* = 0.01) in the border area (Figure 2B). However, no significant change in the protein levels of either TNKS was observed in the remote area (Figure 2A,B). Surprisingly, the protein level of AXIN1 was increased in the border area at 1 week post-MI (border vs. sham, *p* = 0.01; border vs. remote, *p* = 0.06) (Figure 2C) when the hearts initiated adaptive response to ischemic injury, but not at 4 weeks post-MI (Figure 2D). MI boosted the level of Act-β-CAT in the border area at 4 weeks (border vs. sham, *p* = 0.02) but had no significant effect at 1 week post-MI (Figure 2C,D). The protein level of full-length DICER showed significant increase in the border area compared to sham-operated controls at 1 week post-MI (border vs. sham, *p* = 0.009) (Figure 2E), whereas the ΔDICER was elevated in the border area at both 1 week (border vs. sham, *p* = 0.002; border vs. remote, *p* = 0.01) and 4 weeks (border vs. sham, *p* = 0.01; border vs. remote, *p* = 0.06) (Figure 2E,F). The data indicate that ischemic injury induces the augmentation of TNKSs and DICER in the border area during cardiac remodeling and scar formation.

We further employed another set of MI rats to investigate the protein expression and activity of TNKSs in infarct areas at 2, 4, and 8 weeks post-MI in comparison with sham-operated controls. The MI rats displayed increased left ventricular volumes (Figure 3A,B) with preserved LVEF (Figure 3C) at 8 weeks post-MI. Relative heart weight was higher in the MI rats compared with the sham-operated controls as shown previously [25]. The data indicate MI rats developed cardiac dilatation and hypertrophy at 8 weeks post-MI. We observed a substantial increase in TNKS1 protein levels at all examined time points (2 weeks, *p* = 0.006, median sham = 0.94, infarct = 36.63; 4 weeks, *p* = 0.018, median sham = 0.97, infarct = 12.26; 8 weeks, *p* = 0.024, median sham = 1.01, infarct = 13.20) (Figure 3D–F), accompanied by a significant increase in TNKS2 at 2 (*p* = 0.004, median sham = 0.99, infarct = 1.64) and 4 (*p* = 0.047, median sham = 1.01, infarct = 1.43) weeks post-MI (Figure 3D,E). Immunohistochemical staining revealed an abundant distribution of TNKS1 in non-cardiomyocytes in the infarct area (Figure 4B–D), whereas the expression of TNKS1 was mainly confined to vasculature in sham-operated ventricles (Figure 4A,A’). In the infarct area, a TNKS1 immunofluorescent signal was observed in granulocyte- (Figure 4E), fibroblast- (Figure 4F), and endothelium-like cells (Figure 4G). Next, we co-stained cardiac tissue at 2 weeks post-MI with TNKS1 and vimentin, a marker for fibroblasts and epithelial–mesenchymal transition (EMT) cells [26,27]. The staining revealed TNKS1 immunofluorescent signals in vimentin-positive cells in deteriorated myocardium (Figure 4H) indicate the expression of TNKS1 in fibroblasts in the infarct area. The TNKS1 signal also displayed in the vimentin-positive EMT cells migrating into subepicardial space and myocardium (Figure 4I), and in vimentin-positive vascular endothelial cells in the infarct area (Figure 4I). To confirm the expression of TNKS1 in vascular endothelial cells, we performed a double staining of cardiac tissue at 2 weeks post-MI with TNKS1 and CD34, a marker of vascular endothelial and circulating progenitor cells [28,29]. We observed the TNKS1 immunofluorescent signal in CD34-positive vascular endothelial cells (Figure 4J) and circulating cells (Figure 4K). The results indicate that MI induces an abundant distribution of TNKS1 in a subset of non-cardiomyocytes with enhanced progenitor activity. In contrast to TNKS1, TNKS2 was mainly expressed in cardiomyocytes displaying cell shrinkage in the infarct area (Figure 4M–O), while only a faint TNKS2 immunofluorescent signal was visualized in normal cardiomyocytes in sham-operated ventricles (Figure 4L).

Further, AXIN1 showed a tendency to decline at 4 weeks and was significantly decreased at 8 weeks (*p* = 0.002, median sham = 1.09 and infarct = 0.061) in the infarct area compared to sham controls (Figure 5B,C). In line with the decline of AXIN1, both total and active β-CAT were clearly augmented from 4 weeks onward (4 weeks: total β-CAT, *p* = 0.007, median sham = 1.01 and infarct = 1.94; Act-β-CAT, *p* = 0.004, median sham = 1.00 and infarct = 2.16; 8 weeks: total β-CAT, *p* = 0.01, median sham = 1.0 and infarct = 2.03; Act-β-CAT, *p* = 0.03, median sham = 1.02 and infarct = 2.30) (Figure 5B,C). The full-length DICER was increased in the infarct area at 2 weeks post-MI (*p* = 0.03, median sham = 1.10 and infarct = 4.81) (Figure 5D), while the ΔDICER was enriched in the infarct area at all examined time points (2 weeks, *p* = 0.005, median sham = 0.84 and infarct = 6.68; 4 weeks, *p* = 0.03, median sham = 0.95 and infarct = 10.05; 8 weeks, *p* = 0.05, median sham = 1.00 and infarct = 7.47). Notably, the protein levels of AXIN1 and Act-β-CAT displayed a general trend toward increase at 2 weeks post-MI (Figure 5A), possibly reflecting the cardiac adaptive response to ischemic injury. Collectively, MI promotes the activity of TNKS in the infarct area from 4 weeks onward.

### 2.4. MI Activates Wnt/β-Catenin Signaling in Rat Hearts

The augmentation of Act-β-CAT in infarcted myocardium prompted us to investigate the activation of Wnt/β-catenin signaling. Consistent with the activation of β-CAT, *Axin2,* a target of Wnt/β-catenin signaling, exhibited a trend toward increase in the infarct area at 4 weeks (*p* = 0.055, median sham = 0.96 and infarct = 1.76), and significant increase at 8 weeks post-MI (*p* = 0.03, median sham = 1.02 and infarct = 3.10) in comparison with sham controls (Figure 5G). We further investigated the possible effect of Wnt protein on the activation of Wnt/β-catenin signaling. Intriguingly, *Wnt4*, which had previously shown a strong upregulation in mouse adult hearts [30] and a slight increase in neonatal hearts [31] after cardiac injury, was downregulated in infarcted myocardium at 8 weeks post-MI (*p* = 0.02, median sham = 1.01 and infarct = 0.733) (Figure 5H). This implies that activation of Wnt/β-catenin signaling in response to ischemic injury might be associated with the reduction in AXIN1 due to the activation of TNKSs.

### 2.5. Tnks1 and Tnks2 Display Distinct Expression Pattern in Zebrafish Developing and Adult Heart

Zebrafish has emerged as a compelling vertebrate model for defining the molecular basis of cardiac disease and accessing the therapeutic potential of small molecules due to several advantages over traditional mammalian models [32]. We wanted to explore whether inhibition of TNKS activity could prevent cardiac remodeling and protect against cardiac dysfunction using a zebrafish IHF model. To achieve this, we first detected the expression pattern of Tnks1 and Tnks2 in zebrafish heart. In the developing heart, Tnks1 was widely distributed in the ventricle, atrium, and bulbus arteriosus (Figure 6A–C), whereas Tnks2 expression was confined to the atrium (Figure 6D–F). Furthermore, whole-mount immunofluorescence analysis of 4 dpf larval heart revealed that Tnks1 was co- expressed with Mhc in the myocardium (Figure 6C), whereas Tnks2 was co-localized with Raldh2, an epicardial marker, in the epicardium with stronger co-localization in the atrium (Figure 6F). In the adult heart, Tnks1 expression was restricted to the coronary vasculature in the cortical myocardium and blood cells in the lumen (Figure 6G–I), while Tnks2 was localized in the epicardium and myocardium (Figure 6J–L). In the adult ventricular myocardium, Tnks2 predominated in the nuclei of cardiomyocytes and the coronary vasculature (Figure 6J). Collectively, Tnks1 and Tnks2 exhibit different spatiotemporal expression patterns in the developing and adult zebrafish heart, implying a possible functional role of Tnks1 in cardiac development and Tnks2 in cellular homeostasis.

### 2.6. Tnks Inhibition Protects against ISO-Induced Cardiac Dysfunction in Adult Zebrafish

Abnormal β-adrenergic receptor (β-AR) signaling is common in human cardiac disease and contributes significantly to HF. Acute or chronic β-AR activation by ISO induces cardiac remodeling and HF in rodents similar to MI in humans; hence, it represents an established IHF model [33,34]. Accordingly, we investigated whether treatment with a high dosage of ISO could induce cardiac dysfunction in adult zebrafish similar to mammals. Indeed, ISO treatment led to systolic dysfunction exemplified by marked decrease in the ejection fraction (EF) (vehicle vs. ISO, *p* = 0.008), fractional shortening (FS) (vehicle vs. ISO, *p* < 0.001), and fractional area change (FAC) (vehicle vs. ISO, *p* < 0.001) (Figure 7A–C). In addition, ISO treatment resulted in ventricular dilatation (Figure 7D–E), as shown by the significantly enlarged end-systolic ventricle area (ESA) (vehicle vs. ISO, *p* = 0.003) (Figure 7G) and end-diastolic ventricle area (EDA) (vehicle vs. ISO, *p* = 0.02) (Figure 7H). No difference in heart rate (HR) was observed in ISO-treated zebrafish in comparison with controls (Figure 7I). Our data suggest that exposure of adult zebrafish to high dosage of ISO substantially resulted in cardiac dysfunction and dilatation after seven days. Importantly, cardiac dysfunction was ameliorated by inhibition of Tnks activity, as administration of XAV939 prevented the decline of EF, FS, and FAC (Figure 7A–C) and the enlargement of ESA (Figure 7D,F,G) and EDA (Figure 7H) in ISO-treated zebrafish. This was demonstrated by no significant difference in these cardiac parameters between zebrafish treated with vehicle or ISO together with XAV939. Taken together, inhibition of Tnks activity protected against ISO-induced cardiac dysfunction and ventricular dilatation, suggestive of a cardioprotective effect of TNKS inhibition shortly after cardiac injury.

### 2.7. Tnks Inhibition Abrogates ISO-Induced Activation of Wnt Signaling in Adult Zebrafish Hearts

We next analyzed the activation of cardiac Wnt/β-catenin signaling in ISO-treated zebrafish upon Tnks inhibition. ISO treatment boosted Wnt signaling, illustrated by the upregulation of *axin2* (vehicle vs. ISO *p* = 0.03) and *wnt4* (vehicle vs. ISO *p* = 0.008) (Figure 8A). Administration of XAV939 to ISO-treated zebrafish alleviated the upregulation of *axin2,* but not *wnt4* (vehicle vs. ISO+XAV *p* = 0.03) (Figure 8A). To further confirm the XAV939-mediated effect on *axin2,* we analyzed the cardiac expression of *axin2* and *wnt4* in zebrafish treated with XAV939 alone in comparison with those treated with the vehicle. As expected, XAV939 treatment downregulated the expression of *axin2* (vehicle vs. XAV *p* = 0.03) but had no effect on the expression of *wnt4* (Figure 8B). The results demonstrate that administration of XAV939 to ISO-treated zebrafish abrogates ISO-induced activation of Wnt/β-catenin signaling via inhibiting Tnks activity.

Wnt/β-catenin signaling regulates fibrosis and hence affects collagen-rich scar size after MI [17]. To investigate whether activation of Wnt signaling induced by ISO promotes fibrotic deposition, we analyzed the expression of *col1a1a*, the most abundant cardiac collagen gene, and *col5a1,* a minor constituent of scar tissue in zebrafish hearts. The expression of both *col1a1a* and *col5a1* displayed a tendency toward increase in ISO-treated hearts, whereas administration of XAV939 to ISO-treated zebrafish attenuated the trend of increase in both genes (Figure 8C). The results suggest that TNKS inhibition with XAV939 likely protects from cardiac fibrosis upon cardiac injury.

### 2.8. Tnks Inhibition Prevents ISO-Induced Dysregulation of miRNAs in Adult Zebrafish Hearts

Given that TNKS2 and miR-34a-5p and miR-21-5p were elevated in the LV of IHF patients, we investigated the impact of Tnks inhibition on the expression of these two miRNAs in ISO-treated zebrafish hearts. Interestingly, we found that ISO treatment reduced the expression of miR-34a-5p (vehicle vs. ISO, *p* = 0.009) (Figure 8D). Importantly, administration of XAV939 amended ISO-induced downregulation of miR-34a-5p (Figure 8D). Cardiac expression of miR-21-5p, however, remained unchanged in zebrafish either treated with ISO alone or together with XAV939 (Figure 8D). We further investigated the cardiac expression of these two miRNAs in zebrafish treated either with XAV939 or the vehicle. XAV939 treatment resulted in the upregulation of miR-34a-5p (vehicle vs. XAV, *p* = 0.03), while the expression of miR-21-5p showed no significant difference between XAV939-treated hearts and controls (Figure 8E). Taken together, the findings indicate that Tnks inhibition prevents ISO-induced downregulation of miR-34a-5p, and Tnkss have potential impact on the regulation of miRNAs.

## 3. Discussion

In the present study, we reveal enhanced protein levels and activity of TNKSs in the mammalian myocardium in response to ischemic injury. Furthermore, these two TNKSs display differential temporal and spatial expression of protein levels at different phases of the disease progression. Specifically, both TNKS1 and TNKS2 were elevated in the infarct and border area in rat hearts undergoing cardiac remodeling and scar formation, whereas TNKS1 was continually increased in the infarct area in rats developing cardiac hypertrophy. Infarct expansion and extension are two major mechanisms that contribute to adverse left ventricular remodeling following MI. Remodeling of myocardium in the infarct and border areas triggers signaling pathways that can initiate a myopathic process involving the left ventricle globally, progressively enlarging the infarct area and worsening cardiac function [35,36]. Accordingly, activation of TNKSs in both the infarct and border areas might play a role in ventricular remodeling. Moreover, TNKSs appear differently regulated in rat MI vs. IHF patients’ hearts. In rat MI hearts, neither TNKS1 nor TNKS2 was increased in the remote area, whereas in end-stage IHF hearts, TNKS2 was augmented in non-infarcted myocardium. This might be ascribed to end-stage IHF patients developing global adverse LV remodeling while rats have remodeling in the infarct and surrounding myocardium even at 8 weeks post-MI. In addition, the observation of elevated TNKS1 in MI rats and TNKS2 in patients’ LVs may indicate a major functional role of TNKS1 in rat heart and TNKS2 in human heart. The fact that Tnks2, but not Tnks1, abundantly expresses in adult zebrafish epicardium and myocardium together with highly conserved cardiac action potentials and contractile dynamics in zebrafish with humans over mice make zebrafish a compelling HF model for studying the pharmacotherapeutic potential of TNKS inhibition for prevention of IHF.

Our results are in line with previous studies showing PARP activation in human failing hearts [37,38,39]. Identifying potential PARP members associated with cardiac remodeling and heart failure has mainly focused on PARP1, the most abundant PARP protein. A prior study ascribed PARP activation in human failing hearts to markedly increased PARP1 expression [38]. To the contrary, PARP activation appeared unrelated to PARP1 in another study showing comparable PARP1 expression between failing and donor hearts [39]. The discrepancy raises a question whether PARP1 accounts for the entire PARP overactivation in human failing hearts. Intriguingly, our study unveils TNKSs as potential PARP members contributing to PARP activation in failing hearts. In fact, *TNKS* appeared to be associated with hypertension by genome-wide association meta-analysis of systolic and diastolic blood pressure in a large population across five ancestries [23]. The association of *TNKS* and *TNKS2* with CVD and ischemic stroke risk has also been identified by a whole genome survey of Caucasian women [22]. These data together illustrate a critical role of TNKSs in the pathogenesis of cardiac and vascular dysfunction.

A key finding in this study is that TNKS inhibition with a selective TNKS1/2 inhibitor, XAV939, protected against cardiac dilatation and dysfunction provoked by ISO in zebrafish. Supporting our finding, mice additionally exposed to a combination of SB431542 (transforming growth factor-β inhibitor) and XAV939 after MI displayed improved cardiac function compared with those exposed to only cardiac transcription factors (Gata4, Mef2c, and Tbx5) [40]. Moreover, inhibition of PARylation activity with 3-aminobenzamide (3-AB) significantly improved cardiac function in MI rats [41]. In line with this, PARP inhibition with L-2286 preserved systolic LV function and attenuated cardiac hypertrophy and interstitial fibrosis in ISO-induced chronic heart failure in rats [42]. Both 3-AB and L-2286 are unselective inhibitors of multiple PARP family members [43], which may limit their potential for clinical applications. Nevertheless, these studies suggest that PARP inactivation provides a potential approach to protect against cardiac dysfunction in heart failure. We inventively explored the cardioprotective effect of selective TNKS inhibition by XAV939 in an ISO-induced zebrafish HF model. The therapeutic potential of TNKS inhibition has been widely studied in several types of cancer, but the cardiovascular potential of TNKS inhibition has remained unexplored. Thus, our work opens a new avenue proposing that TNKS inhibitors could be a potential therapeutic option to prevent development of HF.

We observed that Wnt/β-catenin signaling is activated in the infarcted myocardium concomitant with TNKS1 augmentation and AXIN1 destabilization in rats. Activation of Wnt/β-catenin signaling was also found in ISO-induced failing hearts in zebrafish. Reactivation of this pathway is well-demonstrated in response to cardiac injury [16,44], suggesting that Wnt/β-catenin signaling is a potential target for therapeutic intervention. In a previous study, oral administration of pyrvinium, a Wnt signaling inhibitor, reduced fibrosis and scar formation and improved EF after MI induced by permanent coronary artery occlusion in mice [45]. In contrast, interruption of Wnt/β-catenin signaling in cardiac fibroblasts or epicardial cells leads to cardiac dysfunction after acute cardiac injury [14]. The discrepancy could be due to the type of experimental MI model, acute vs. chronic injury, and the approach to the interruption of Wnt/β-catenin signaling, systemic inhibition or cardiac cell type-specific disruption. In the present study, administration of XAV939 attenuated Wnt/β-catenin signaling and protected against cardiac dysfunction in ISO-treated zebrafish. We assume that TNKS inhibition may exert cardioprotective effects partially via suppression of Wnt/β-catenin signaling mediated by AXIN1.

Notably, we found that TNKS2 and DICER increased along with the upregulation of miR-34a-5p and miR-21-5p in the non-infarcted myocardium in IHF patients. We also observed heightened TNKS2 and DICER in the border and infarct areas at the early phase, but not at the late phase when TNKS1 remained elevated in MI rats. TNKS inhibition amended ISO-induced dysregulation of miR-34a-5p in zebrafish. Thus, our study raises an interesting hypothesis that TNKS2 may modulate miRNAs via DICER.

We found that miR-34a-5p was upregulated in the LV of IHF patients but downregulated in ISO-treated zebrafish hearts. Similarly, myocardial expression of miR-34a was markedly elevated in adult mice after MI, while it was reduced in neonatal mice at 7 days post-MI [46,47]. A prior study indicated that miR-34a dampens growth factor signaling for cell survival and proliferation and key genes regulating cell cycle progression [48]. Thus, the expression level of miR-34a is negatively correlated with cardiac regenerative capacity. Whether downregulation of miR-34a-5p exerted a detrimental or compensatory effect in response to ISO treatment in zebrafish hearts in the present study needs to be clarified in further studies.

Interestingly, we found that miR-21-5p was upregulated in the LV of IHF patients. In line with our finding, increased expression of miR-21 has been observed in the LV of end-stage heart failure patients [49] and in the left atria of patients with atrial fibrillation [50]. In addition, elevated miR-21 promoted cardiac fibrosis and dysfunction in a mouse pressure-overload model [51]. Consequently, inhibition of miR-21 with antimir-21 prevented cardiac fibrosis and hypertrophy and improved cardiac function in a mouse pressure-overload model [51] and in a pig model of I/R injury [52], implying a role for miR-21 in cardiac fibrosis and hypertrophy. We detected unaltered miR-21-5p in zebrafish hearts in response to ISO treatment in the present study. The discrepancy might be ascribed to the different phases of cardiac remodeling and pathological progression in IHF patients and the ISO-induced zebrafish IHF model.

In conclusion, our data indicate that the abundance and activity of TNKSs are elevated in the myocardium of IHF patients and MI rats. We demonstrate that pharmacological inhibition of TNKS activity prevents cardiac dilatation and dysfunction in zebrafish, possibly through regulation of miRNA expression and suppression of Wnt/β-catenin signaling via DICER and AXIN1, respectively. These data highlight the pharmacotherapeutic potential of TNKS inhibition shortly after cardiac injury for prevention of IHF. Such potential could be confirmed in an in vivo mammalian model to gain more translational insight into pharmacological effects of TNKS inhibition in the future.

## 4. Materials and Methods

### 4.1. Human IHF Patients

The LV tissue samples were obtained from patients aged 49 to 66 years old and undergoing cardiac transplantation due to end-stage IHF in Helsinki University Hospital. The myocardium samples were taken from the non-infarcted regions and immediately frozen in isopentane solution (2-methylbutane) cooled in liquid nitrogen and stored at −80 °C. The investigation conformed to the Helsinki Declaration. The Ethics Committee of Helsinki and Uusimaa Hospital District approved the study protocol. Written informed consent was received from the patients. Control samples were derived from the LVs of organ donors without cardiac disease, whose hearts could not be used as whole organ grafts due to tissue type or size mismatch. The National Authority for Medicolegal Affairs approved the usage of tissues from organ donors. The age, gender, and medical and medication history of the patients are listed in Table 1.

### 4.2. Rat MI Model

Rat myocardial infarction was induced by ligation of the left anterior descending (LAD) coronary artery as described previously in detail [25,53]. The experimental protocol was approved by the Regional Government Office of Southern Finland (ESHL-2003-07630/Ym-23; ESLH-2009-04300/Ym23). Briefly, male Wistar rats were anesthetized by subcutaneous (s.c.) injection with 0.5 mg/kg of medetomidine (Domitor, Orion, Turku, Finland) and 60 mg/kg of ketamine (Ketalar, Parke Davis, Barcelona, Spain), and intubated for mechanical ventilation. The heart was exposed through a lateral thoracotomy, and the LAD was occluded at three millimeters from its origin. The sham-operated rats underwent the same procedure, except for the ligation of the coronary artery. Post-operatively, rats were hydrated with saline (s.c.) and given 0.05 mg/kg buprenorphine (Temgesic^®^, Schering-Plough, Brussels, Belgium) (s.c.) twice daily for three days for analgesia. Rats with an infarct size of under 12% of the left ventricle or only endocardial infarction were excluded from the study. At the end of the experiment, the rats were sacrificed under 55 mg/kg pentobarbital (Orion, Espoo, Finland) anesthesia (i.p.). The hearts were cut into three 2 mm transverse slices below the point where the coronary artery was ligated. Two slices were immediately frozen in liquid nitrogen and stored at −80 °C. One slice was fixed in 10% formalin followed by embedding in paraffin. The sample preparation from heart excision to immersion in liquid nitrogen was completed within 2–3 min. This period of time may alter the intracellular proteins and result in unpredictable changes. However, the myocardial samples from MI rats and sham controls were prepared following the same procedure. Thus, the period of time intervening between the heart cut and the immersion into the liquid nitrogen could affect the intracellular proteins equally in sham-operated and infarcted myocardial samples. Tissue samples from two previous experimental sets [25,53] were employed in this study.

Experimental set 1: rats were killed at one week or four weeks after LAD ligation or sham operation [53]. Experimental set 2: this experimental set was initially designed to study the effect of a synthetic carbon monoxide releasing molecule CORM-3 on cardiac recovery after MI; therefore, the control groups of both sham-operated and infarcted hearts received inactive CORM-3 (iCORM-3, 20 mg/kg i.p.) from day 4 to 14 after MI. In the present study, we employed iCORM-3-treated control groups. Considering that iCORM-3 could equally affect cardiac function in sham-operated and infarcted hearts, the myocardial structural and functional difference between the two groups can be associated with MI. Rats were killed at two, four or eight weeks after MI or sham operation [25].

### 4.3. Zebrafish IHF Model and TNKS Inhibitor Treatment

The National Animal Experiment Board in Finland has granted animal permits for execution with zebrafish (ESAVI/4131/04.10.07/2017; ESAVI/16286/2020). Adult zebrafish were obtained from the breeding line of a Turku strain. They were maintained in a recirculating aquatic system in the zebrafish core facility at the University of Helsinki. To induce HF, fish were anesthetized with 0.02% tricaine in system water and i.p. injected with a single high dose of isoproterenol (ISO, Sigma-Aldrich, Munich, Germany) at 150 mg/kg body weight [33]. A TNKS-specific inhibitor, XAV939 (Selleckchem, Munich, Germany), that has a 200-fold selectivity over PARP1 and PARP2 [54] was administrated through fish water at a final concentration of 10 μM from one day post-injection (1 dpi) for seven days [20]. Fish were randomly divided into three groups: control, ISO, and ISO+XAV939. Control zebrafish received saline. To investigate the effect of XAV939 on healthy zebrafish, fish were randomly divided into two groups: one group received DMSO as control and another received XAV939. Administering 10 μM of XAV939 to zebrafish had no effect on viability, swimming activity, or obvious morphological phenotype. At 7 dpi, echocardiographic examination was conducted. At 8 dpi, fish were anesthetized with 0.02% tricaine in system water and hearts were dissected. Three individual hearts were pooled and snap-frozen in dry ice and stored at −80 °C.

### 4.4. Echocardiography

Echocardiography of rats was performed with a 7.5 MHz transducer (MyLabR25, Esaote SpA, Genoa, Italy) as described previously [25]. Rats were sedated with 0.5 mg/kg medetomidine. Images were obtained from the left parasternal short-axis views of the left ventricle (LV). Anterior and posterior wall thicknesses and internal diameters during systole (LVESD) and diastole (LVEDD) were measured from three cardiac cycles. Left ventricular end diastolic volume (LVEDV), left ventricular end systolic volume (LVESV), and left ventricular ejection fraction (LVEF) were calculated using LVEDD and LVESD by the following formulas: LVEDV = 1.047(LVEDD)^3^; LVESV = 1.047(LVESD)^3^; LVEF = (LVEDV-LVESV)/LVEDV*100 as described previously [55].

Echocardiography of zebrafish was performed using a Vevo 2100^®^ Image System and Vevo Imaging Station (VisualSonics, Amsterdam, The Netherlands) equipped with a high-frequency transducer (MS700, 30–70 MHz) as described previously [56]. In brief, fish were anesthetized with 0.02% tricaine in system water. Echocardiographic images were acquired in the longitudinal axis (LAX) view to record B-mode videos and in the short axis (SAX) view to obtain pulsed-wave Doppler (PWD) signals. Image analysis was conducted offline using Vevo Lab™ analysis software (VisualSonics) by experienced personnel in a blinded manner as described recently [57]. Briefly, ventricular longitudinal diameters from five consecutive cardiac cycles were obtained from B-mode images by measuring the perpendicular distance from the base to the apex at end-diastole (EDD) and end-systole (ESD). Ventricular area at end-diastole (EDA) and end-systole (ESA) were defined as the area within the inner border of the compact myocardium. Diastolic and systolic ventricular volumes (EDV and ESV) were then calculated based on the corresponding diameters and areas using the formula, EDV = 8 * EDA^2^/3π * EDD or ESV = 8 * ESA^2^/3π * ESD. Fractional shortening (FS), fractional area change (FAC), and ejection fraction (EF) were determined from systolic and diastolic ventricular diameter, area, and volume, respectively. For assessment of diastolic function, maximal velocity of blood inflow across the atrioventricular valve during early diastole (E wave) and during atrial systole (A wave) was derived from PWD signals. Heart rates were also obtained from PWD images.

### 4.5. Western Blotting

Human LV tissues were homogenized in ice-cold PBS containing phosphatase inhibitors (Roche Diagnostics GmbH, Mannheim, Germany) and protease inhibitor cocktail (EDTA-free, Roche) with a Precelly^®^ CK28 lysing kit (Bertin Corp., Rockville, MD, USA) using a homogenizer (Precellys, Bertin Corp.). Ice-cold n-Dodecyl-β-D-maltoside (1%) was immediately added to the lysates. Rat heart tissues were homogenized in a chilled lysis buffer (100 mM NaCl, 10 mM KCl, 8 mM Na_2_HPO_4_, 3 mM MgCl_2_, 0.5% NP-40, 10 mM Tris, pH 7.4) supplemented with a protease inhibitor cocktail (Roche) using a Tissue-Tearor (BioSpec, Bartlesville, OK, USA). Following incubation on ice for 30 min, the lysates were pelleted at 14,000× *g* for 15 min at 4 °C. Protein extracts were resolved on 7.5–10% SDS-PAGE and transferred to PVDF membranes (Bio-Rad Laboratories, Hercules, CA, USA). The membranes were incubated with the primary antibodies followed by horseradish peroxidase-conjugated secondary antibodies. The blots were detected using Pierce™ Enhanced Chemiluminescence (ECL) substrate (Pierce Biotechnology, Rockford, IL, USA). Quantification was performed using a Gel Doc Image Analyzer (Bio-Rad). Western blotting was performed at least three times. Antibodies are listed in Table 2.

### 4.6. RNA Extraction and Real-Time Quantitative RT-PCR

Owing to the conservation of miRNA gene families among species, we employed human miRNA primers to analyze homologous miRNA genes in zebrafish. Sequence search against the miRBase database (http://www.mirbase.org) (accessed on 16 September 2019) using the primers as queries identified the respective homologous miRNAs in zebrafish. To analyze miRNA expression, RNA was extracted from human LV tissue samples with a Maxwell^®^ RSC miRNA Tissue Kit using a Maxwell^®^ Instrument according to the manufacturer’s instructions and from zebrafish hearts with miRNeasy Mini Kit (Qiagen, Hilden, Germany). Reverse transcription was performed with a miScript II RT kit (Qiagen). Quantitative PCR was performed with a miScript primer assay (Qiagen) and miScript SYBR Green PCR kit (Qiagen) using Light Cycler 480 II instrument (Roche Applied Science, Penzberg, Germany). A Small Nucleolar RNA, C/D Box 61 (SNORD 61-11), served as an internal control for normalization.

To detect gene expression, total RNA was extracted from rat myocardium using a mirVana miRNA Isolation Kit (Ambion, Carlsbad, CA, USA) and from dissected zebrafish adult hearts using a miRNeasy Mini Kit (Qiagen) according to the manufacturer’s instructions. Reverse transcription was carried out using a SuperScript^®^ VILO™ cDNA synthesis kit (Invitrogen, Carlbad, CA, USA). Quantitative PCR was performed with gene specific primers (Table 3) and the LightCycler^®^ 480 SYBR Green I Master (Roche). Glyceraldehyde 3-phosphate dehydrogenase (*Gapdh*) and cytochrome P450 A (*CypA*) served as references for normalization in rats, and *gapdh* and elongation factor 1α 1a (*ef1α1a*) in zebrafish. RT-qPCR was performed at least three times with three technical replicates for each sample. Primers are listed in Table 3.

### 4.7. Immunohistochemistry

Paraffin sections from the midventricular level of rat hearts [25] were blocked with CAS-block (Zymed, South San Francisco, CA, USA) and stained with primary antibody for TNKS1, TNKS2, Myosin heavy chain (MHC), CD34, or vimentin diluted in Dako REAL Antibody Diluent (Dako, Glostrup, Denmark) followed by incubation with AlexaFluor-594 and AlexaFluor 488 conjugated secondary antibodies. Nuclei were labeled with DAPI (4′,6-Diamidino-2-Phenylindole) (Molecular Probes, RRID:AB_2307445, Eugene, OR, USA). Sections were mounted with a ProLong Diamond antifade mountant (Molecular Probes) and imaged with a Zeiss LSM 780 confocal microscope (Carl Zeiss Microscopy GmbH, Jena, Germany).

Dissected hearts from four days post-fertilization (4 dpf) larvae were fixed in 4% paraformaldehyde in PBS for 20 min at RT, followed by incubation with TNKS1 and MHC or TNKS2 and retinaldehyde dehydrogenase 2 (RALDH2) antibodies overnight at 4 °C as previously described [61]. The immunoreactivity was detected with AlexaFluor 488 and AlexaFluor 594 labeled secondary antibodies. Images were captured with the Zeiss LSM 780 confocal microscope.

A transgenic line *Tg*(*cmlc2*:*GFP*) that expresses GFP under the control of the heart-specific promoter of cardiac myosin light chain 2 (cmlc2) was generated by injection of a CRISPR vector p*cmlc2:GFP*-*U6:gRNA-Prom:Cas9* (kindly provided by Leonard Zon) along with *Tol2 transposase* mRNA into one-cell embryos [62]. Larvae with specific fluorescence expression in the heart were raised to adulthood and outcrossed to the WT strain to produce a stable transgenic line. Dissected adult hearts from the F3 generation were fixed in 4% PFA followed by sucrose cryoprotecting and embedding in a Tissue-Tek^®^ OCT compound (Sakura, Alphen aan den Rijn, The Netherlands). Sections (8 µm) were blocked with CAS-block (Zymed) and stained with TNKS1 or TNKS2 antibodies followed by incubation with AlexaFluor-594-conjugated secondary antibodies. Nuclei were labeled with DAPI. Sections were mounted with a ProLong Diamond antifade mountant (Molecular Probes) and imaged with the Zeiss LSM 780 confocal microscope. Antibodies are listed in Table 2.

### 4.8. Statistical Analysis

Data were analyzed using Prism 6.0 software (GraphPad, San Diego, CA, USA) and presented as mean ± SD. One-way ANOVA with Tukey adjustment for multiple comparisons was used to calculate differences between more than two groups. A two-sample unpaired two-tailed *t*-test was performed for comparisons between two groups. A *p* value of <0.05 was assigned to be statistically significant.

## Figures and Tables

**Figure 1 ijms-23-10059-f001:**
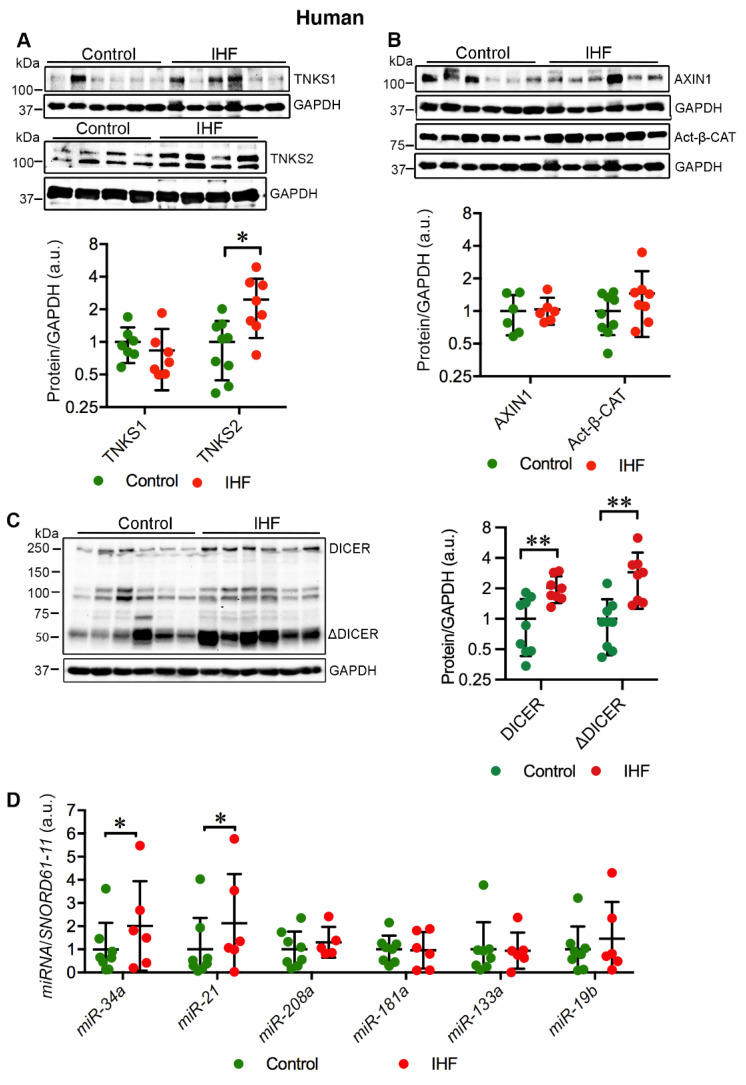
Cardiac TNKS2 and DICER are stabilized in ischemic heart failure (IHF) patients. (**A**) Representative immunoblots of TNKS1 and TNKS2 in the left ventricle (LV) of IHF and healthy controls. (**B**) Representative immunoblots of AXIN1 and active β-catenin (Act-β-CAT) in the LV of IHF and healthy controls. (**C**) Representative immunoblots of DICER and truncated DICER (ΔDICER) in the LV of IHF and healthy controls. The graphs represent quantifications of three replicate blots for the indicated proteins relative to GAPDH. The results are presented as fold change compared to controls. The blurry or oversaturated bands were excluded from the quantification. Controls: TNKS1, *n* = 7; TNKS2, *n* = 9; AXIN1, *n* = 6; Act-β-CAT, *n* = 9; DICER, *n* = 9; ΔDICER, *n* = 9; IHF: TNKS1, *n* = 7; TNKS2, *n* = 8; AXIN1, *n* = 6; Act-β-CAT, *n* = 8; DICER, *n* = 8; ΔDICER, *n* = 8. (**D**) qRT-PCR analysis of miRNA expression in the LV of IHF patients (*n* = 6) and healthy controls (*n* = 8). Data are presented as mean ± SD. Two-sample *t*-test: * *p* < 0.05, ** *p* < 0.01.

**Figure 2 ijms-23-10059-f002:**
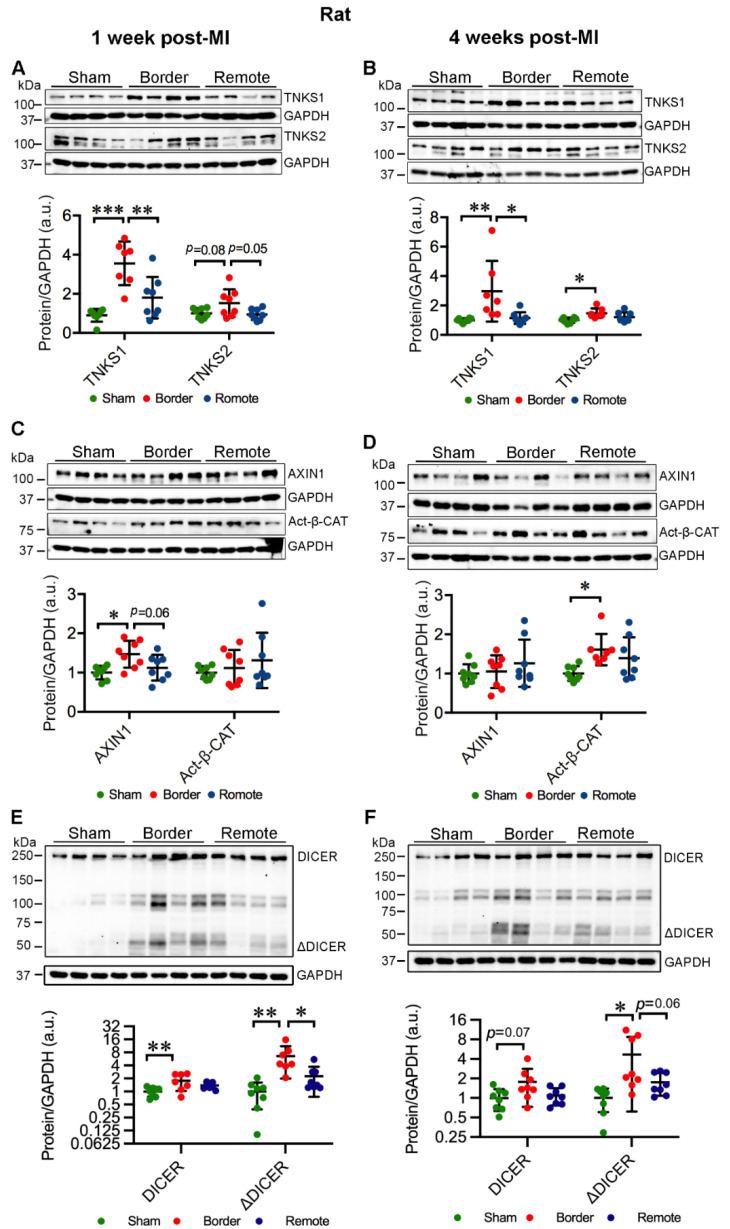
Ischemic injury augments TNKS1 and TNKS2 in the border area in rat hearts. (**A**,**B**) Representative immunoblots of TNKS1 and TNKS2 in the remote and border areas in MI rat hearts and sham-operated controls at 1 week (**A**) and 4 weeks (**B**) post-MI. (**A**) Sham: *n* = 8; MI Border: TNKS1, *n* = 7; TNKS2, *n* = 8; MI Remote: *n* = 8. (**B**) Sham: *n* = 8; MI Border: *n* = 7; MI Remote: *n* = 8. (**C**,**D**) Representative immunoblots of AXIN1 and Act-β-CAT in the remote and border areas in MI rat hearts and sham-operated controls at 1 week (**C**) and 4 weeks (**D**) post-MI. (**C**) Sham: *n* = 8; MI Border: *n* = 8; MI Remote: *n* = 8. (**D**) Sham: *n* = 8; MI Border: AXIN1, *n* = 8; Act-β-CAT, *n* = 7; MI Remote, *n* = 8. (**E**,**F**) Representative immunoblots of DICER and ΔDICER in the remote and border areas in MI rat hearts and sham-operated controls at 1 week (**E**) and 4 weeks (**F**) post-MI. (**E**,**F**): Sham, *n* = 8; MI Border, *n* = 8; MI Remote, *n* = 8. The graphs represent quantifications of three replicate blots for the indicated proteins relative to GAPDH. The results are presented as fold change compared to controls. The blurry or oversaturated bands were excluded from the quantification. Data are presented as mean ± SD. One-way ANOVA with Tukey adjustment for multiple comparisons: * *p* < 0.05, ** *p* < 0.01, *** *p* < 0.001.

**Figure 3 ijms-23-10059-f003:**
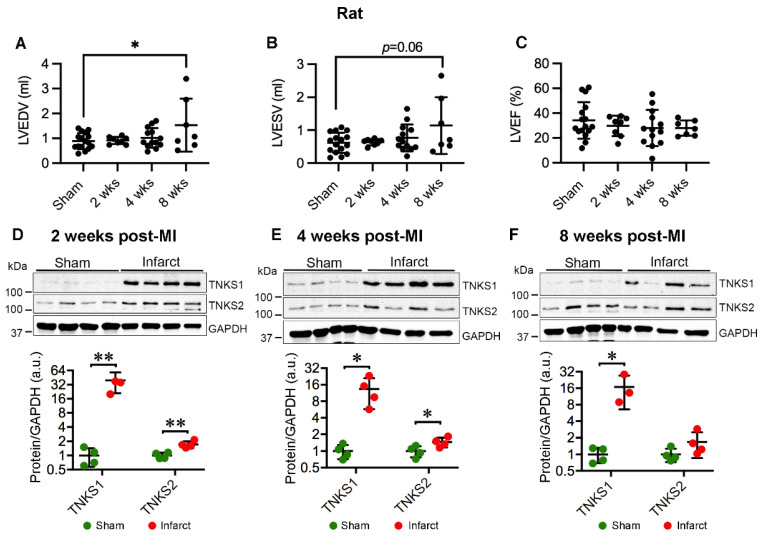
Ischemic injury boosts TNKS1 and TNKS2 in the infarct area in rat hearts. (**A**,**B**) Echocardiography depicts increased left ventricular volume at 8 weeks post-MI compared to sham controls as determined by left ventricular end diastolic volume (LVEDV) (**A**) and left ventricular end systolic volume (LVESV) (**B**). (**C**) Left ventricular ejection fraction (LVEF). (**D**–**F**) Representative immunoblots of TNKS1 and TNKS2 in the infarct area in MI rat hearts and sham-operated controls at 2 (**D**), 4 (**E**), and 8 (**F**) weeks post-MI. The graphs represent quantifications of three replicate blots for the indicated proteins normalized to GAPDH. The results are presented as fold change compared to controls. The blurry band was excluded from the quantification. (**A**–**C**) Sham, *n* = 16; 2 weeks, *n* = 8; 4 weeks, *n* = 13; 8 weeks, *n* = 7. (**D**), Sham, *n* = 4; MI, *n* = 4. (**E**) Sham, *n* = 4; MI, *n* = 4. (**F**) Sham, *n* = 4; MI TNKS1, *n* = 3; TNKS2, *n* = 4. Data are presented as mean ± SD. One-way ANOVA with Tukey adjustment for multiple comparisons (**A**–**C**); two-sample t-test (**D**–**F**); * *p* < 0.05, ** *p* < 0.01.

**Figure 4 ijms-23-10059-f004:**
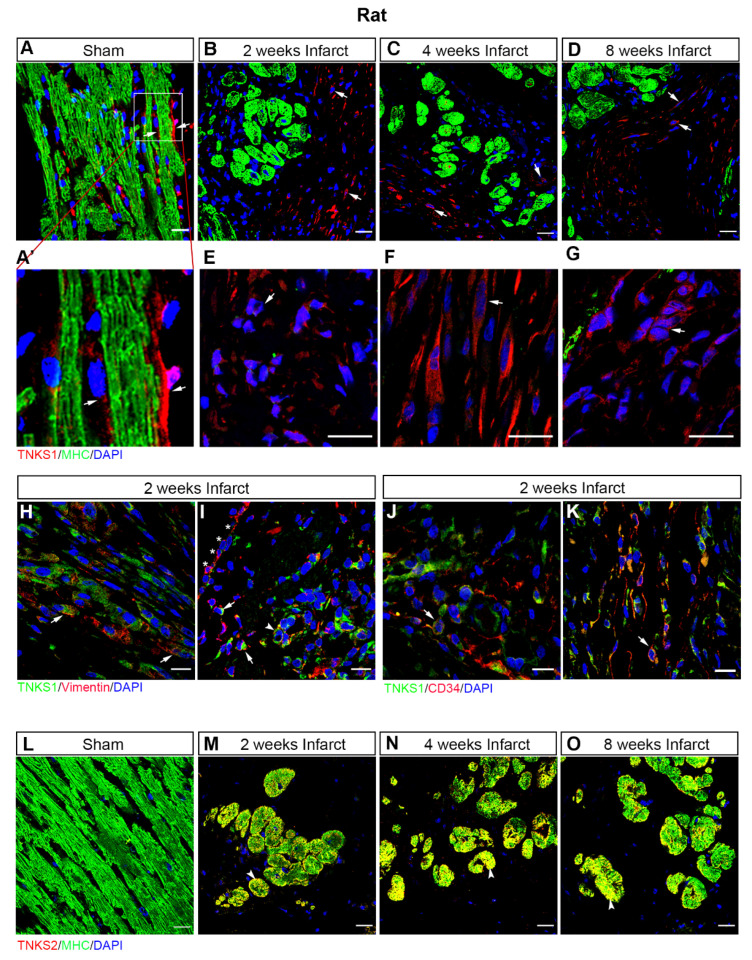
Ischemic injury stimulates TNKS1 in non-cardiomyocytes and TNKS2 in cardiomyocytes in the infarct area in rat hearts. (**A**–**D**) Representative immunohistochemical staining of TNKS1 (in red) and MHC (in green) in ventricular sections from sham-operated and MI rat hearts at 2, 4, and 8 weeks post-MI. The TNKS1-positive signal is visualized in the vasculature (arrow) in sham-operated hearts (**A**,**A’**) and in non-cardiomyocytes (examples indicated by arrows) in the infarct area at the examined time points (**B**–**D**). (**A’**) *Inset*: magnified image of TNKS1-positive cells within the vasculature. (**E**–**G**) Representative images of immunohistochemical staining showing the TNKS1-positive signal in granulocytes-like cells (**E**), fibroblasts (**F**), and endothelium-like cells (**G**) indicated by the arrow in the infarct area. (**H**,**I**) Representative immunohistochemical staining of TNKS1 (in green) and vimentin (in red) showing accumulation of TNKS1 in vimentin-positive fibroblasts (arrow in **H**), EMT cells (arrow in **I**), and vascular endothelial cells (arrowhead in **I**) in the infarct area at 2 weeks post-MI. Stars (* in I) indicate the epicardium. (**J**,**K**) Representative immunohistochemical staining of TNKS1 (in green) and CD34 (in red) showing accumulation of TNKS1 in CD34-positive (arrow) vascular endothelial cells (**J**) and circulating cells (**K**) in the infarct area at 2 weeks post-MI. (**L**–**O**) Representative immunohistochemical staining of TNKS2 (in red) and MHC (in green) in ventricular sections from sham-operated and infarcted rat hearts at 2, 4, and 8 weeks post-MI. The co-localization of TNKS2 and MHC is visualized in cardiomyocytes (arrowhead) in the infarct area at the examined time points. Scale bars: 20 μm.

**Figure 5 ijms-23-10059-f005:**
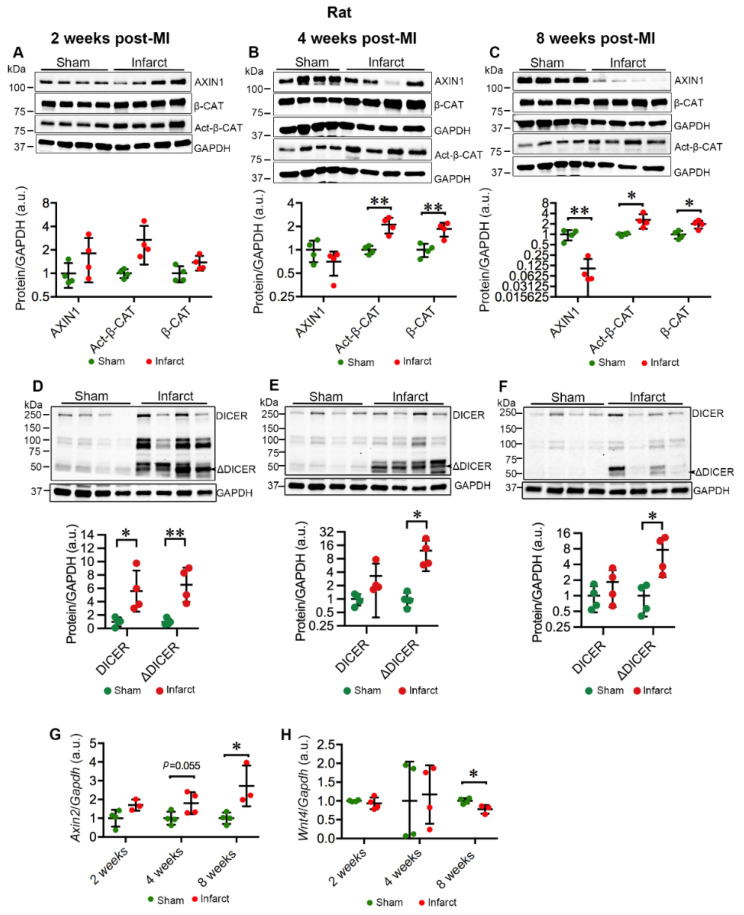
MI destabilizes AXIN1 and activates Wnt/β-catenin signaling in the infarct area in rat hearts. (**A**–**C**) Representative immunoblots of AXIN1, total β-catenin (β-CAT), and active β-catenin (Act-β-CAT) in the infarct area in MI rat hearts and sham-operated controls at 2 (**A**), 4 (**B**), and 8 (**C**) weeks post-MI. (**D**–**F**) Representative immunoblots of DICER in the infarct area in MI rat hearts and sham-operated controls at 2 (**D**), 4 (**E**), and 8 (**F**) weeks post-MI. The graphs represent quantifications of three replicate blots for the indicated proteins normalized to GAPDH. The results are presented as fold change compared to controls. Sham, *n* = 4; MI, *n* = 4. (**G**,**H**) qRT-PCR analysis of *Axin2* (**G**) and *Wnt4* (**H**) in the infarct area of MI rats and sham-operated controls. *Axin2*: Sham, *n* = 4; MI 2 and 8 weeks, *n* = 3; 4 weeks, *n* = 4; *Wnt4*: Sham, *n* = 4; MI 2 and 4 weeks, *n* = 4; 8 weeks, *n* = 3. Data are presented as mean ± SD. Two-sample *t*-test: * *p* < 0.05, ** *p* < 0.01.

**Figure 6 ijms-23-10059-f006:**
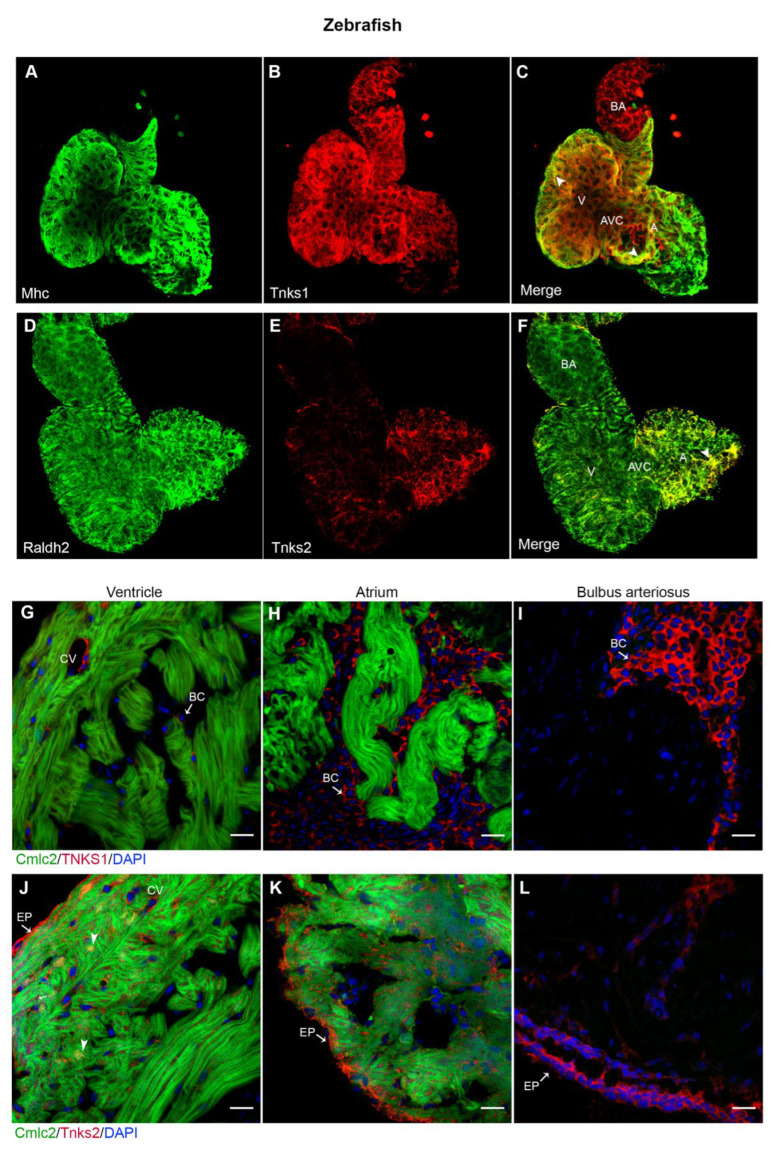
Tnks1 and Tnks2 display a distinct expression pattern in zebrafish developing and adult hearts. (**A**–**C**) Confocal images of whole-mount immunostaining for Mhc (**A**) and Tnks1 (**B**) in dissected zebrafish heart at 4 dpf. Tnks1 is co-expressed with Mhc in the ventricle (arrowhead) and atrium (arrowhead): A, atrium; V, ventricle; BA, bulbus arteriosus; AVC, atrioventricular canal. (**D**–**F**) Representative confocal images of whole-mount immunostaining for Raldh2 (**D**) and Tnks2 (**E**) in dissected zebrafish heart at 4 dpf. Tnks2 is co-expressed with Radldh2 in the epicardium (arrowhead). (**G**–**I**) Representative immunostaining of adult heart sections from transgenic zebrafish *Tg*(*cmlc2*:*GFP*) for Tnks1 (in red) in ventricle (**G**), atrium (**H**), and bulbus arteriosus (**I**). The Tnks1-positive signal is seen in the coronary vasculature (CV) and blood cell (BC) in the lumen. (**J**–**L**) Representative immunostaining of adult heart sections from transgenic zebrafish *Tg*(*cmlc2*:*eGFP*) for Tnks2 (in red) in ventricle (**J**), atrium (**K**), and bulbus arteriosus (**L**). The Tnks2-positive signal is seen in the epicardium (EP), the nucleus of the trabecular and cortical cardiomyocytes (arrowhead), and the coronary vasculature (CV). Scale bars: 20 μm.

**Figure 7 ijms-23-10059-f007:**
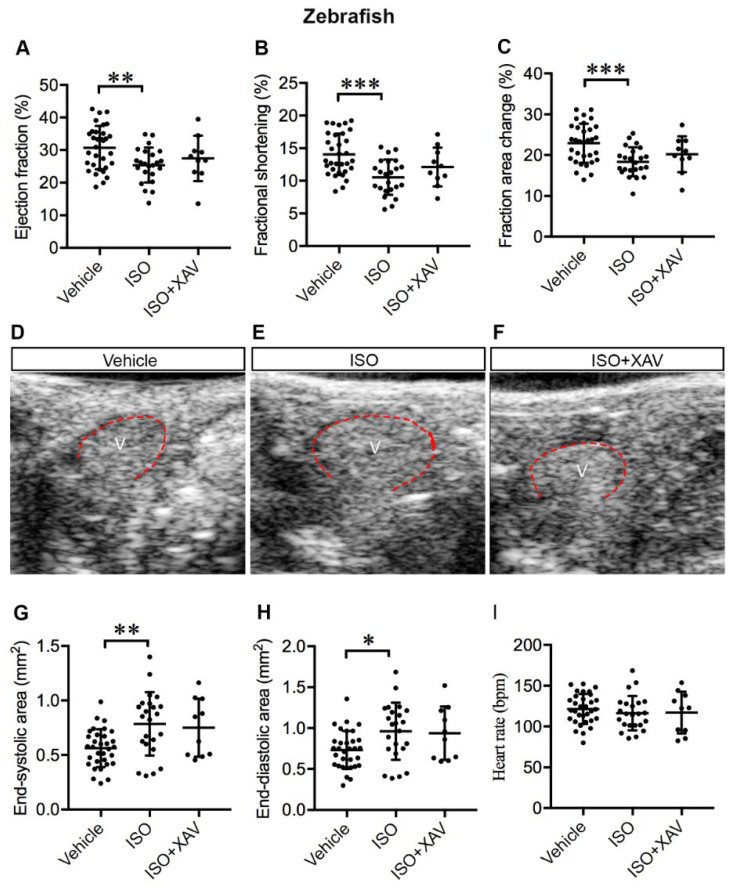
Tnks inhibition protects against cardiac dysfunction and ventricle dilatation induced by ISO in adult zebrafish. (**A**–**C**) Echocardiography depicts restored systolic function upon XAV939 treatment in ISO-treated hearts as determined by the ejection fraction (**A**), fractional shortening (**B**) and fractional area change (**C**). (**D**–**F**) Representative LAX images of adult zebrafish that received vehicle (**D**), ISO (**E**), or ISO and XAV939 (**F**) for 7 days. V: ventricle outlined in red, anterior to the left. (**G**,**H**) XAV939 treatment prevents ventricular dilatation as depicted by the preserved end-systolic (**G**) and end-diastolic (**H**) areas. (**I**) Heart rate displays no significant difference between the groups. Vehicle, *n* = 32; ISO, *n* = 23; ISO+XAV939, *n* = 10. Data are presented as mean ± SD. One-way ANOVA with Tukey adjustment for multiple comparisons: * *p* < 0.05, ** *p* < 0.01, *** *p* < 0.001.

**Figure 8 ijms-23-10059-f008:**
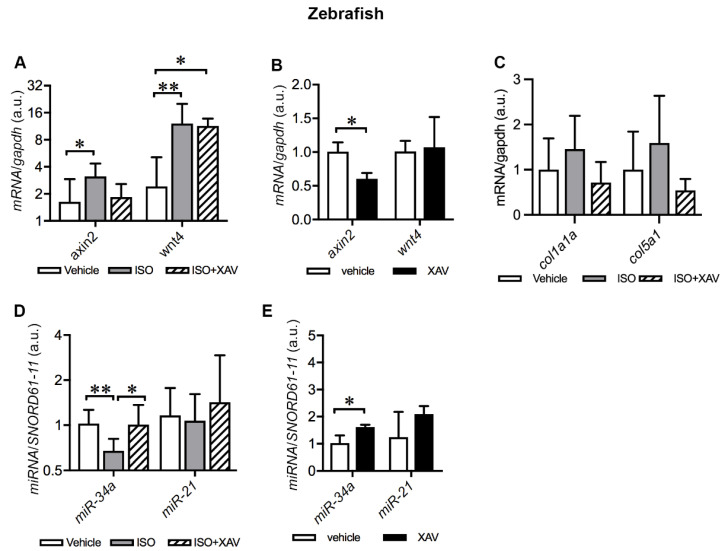
Tnks inhibition amends dysregulation of cardiac Wnt signaling and miRNAs induced by ISO in adult zebrafish. (**A**) qRT-PCR analysis of *axin2* and *wnt4 expression* in zebrafish hearts treated with ISO alone or together with XAV939 compared to the vehicle. (**B**) qRT-PCR analysis of *axin2* and *wnt4 expression* in zebrafish hearts treated with XAV939 compared to the vehicle. (**C**) qRT-PCR analysis of *col1a1a* and *col5a1 expression* in zebrafish hearts treated with ISO alone or together with XAV939 compared to the vehicle. (**D**) qRT-PCR analysis of the expression of selected miRNAs in zebrafish hearts treated with ISO alone or together with XAV9393 compared to the vehicle. (**E**) qRT-PCR analysis of the expression of the selected miRNAs in zebrafish hearts treated with XAV939 compared to the vehicle. The graphs represent the quantification of two individual analyses of RNA extracts from pooling samples of three hearts. Each analysis includes three replicates. (**A**,**C**,**D**): vehicle, *n* = 18 hearts; ISO, *n* = 18 hearts; ISO+XAV939, *n* = 10 hearts. (**B**,**E**): vehicle, *n* = 10 hearts; XAV939, *n* = 10 hearts. Data are presented as mean ± SD. One-way ANOVA with Tukey adjustment for multiple comparisons (**A**,**C**,**D**). Two-sample *t*-test (**B**,**E**): * *p* < 0.05, ** *p* < 0.01.

**Table 1 ijms-23-10059-t001:** Clinical features of ischemic heart failure patients.

Clinical Features	Patients
*No. of patients*	8
Mean age (years)	58
Sex (female/male)	1/7
*Past medical history*	
Coronary artery disease	8
Ischemic heart failure	8
*Previous medication*	
ß-blocker, *n* (%)	7 (87.5)
Statins, *n* (%)	7 (87.5)
Angiotensin receptor inhibitors, *n* (%)	5 (62.5)
Angiotensin-converting enzyme inhibitors, *n* (%)	3 (37.5)
Phosphodiesterase type 5 inhibitors, *n* (%)	6 (75)
Diuretics, *n* (%)	7 (87.5)
Antithrombotic therapy, *n* (%)	8 (100)
Proton pump inhibitors, *n* (%)	8 (100)
Calcium, magnesium, potassium intake	7 (87.5)

**Table 2 ijms-23-10059-t002:** List of antibodies used in the study.

Antibody	Host	Manufacturer	Reference
TNKS	Rabbit polyclonal	Abcam^®^	ab86279, RRID:AB_1925488
TNKS2	Goat polyclonal	Abcam^®^	ab85690, RRID:AB_1861366
AXIN1	Rabbit monoclonal	Cell Signaling Technology	#2087, RRID:AB_2274550
Active β-Catenin	Mouse monoclonal	EMD Millipore	#05-665, RRID:AB_309887
β-Catenin	Mouse monoclonal	Sigma-Aldrich	C7207, RRID:AB_476865
DICER	Rabbit monoclonal	Abcam^®^	ab259327
DICER	Rabbit polyclonal	Abcam^®^	ab227518
RALDH2	Rabbit polyclonal	Bioss Antibodies	BS-3676R, RRID:AB_10856810
MHC	Mouse monoclonal	EMD Millipore	05-716, RRID:AB_309930
Vimentin	Mouse monoclonal	Lab Vision	MS-129-P1, RRID:AB_63350
CD34	Goat polyclonal	R&D system	AF4117, RRID:AB_2074613
GAPDH	Rabbit polyclonal	Abcam^®^	Ab9485, RRID:AB_307275
HRP-rabbit IgG	Goat polyclonal	Jackson ImmunoResearch	111-035-003, RRID:AB_2313567
HRP-mouse IgG	Goat polyclonal	Jackson ImmunoResearch	115-035-044, RRID:AB_2338503
HRP-goat IgG	Rabbit polyclonal	Jackson ImmunoResearch	305-035-003, RRID:AB_2339400
Alexa Fluor^®^ 488-mouse IgG	Goat polyclonal	Invitrogen	A-11001, RRID:AB_2534069
Alexa Fluor^®^ 488-rabbit IgG	Goat polyclonal	Invitrogen	A-11008, RRID:AB_143165
AlexaFluor^®^ 594-rabbit IgG	Goat polyclonal	Invitrogen	A-11012, RRID:AB_2534079
AlexaFluor^®^ 546-goat IgG	Donkey polyclonal	Invitrogen	A-11056, RRID:AB_2534103

**Table 3 ijms-23-10059-t003:** List of primers used in the study.

Target	Species	Primer Sequences	Reference
Axin2	Rattus norvegicus	Forward 5′-CAGCAAAACTCTCCGGGCCA-3′ Reverse 5′-GCGTCGCTGGATAACTCGCT-3′	This study
Wnt4	Rattus norvegicus	Forward 5′-GGACAGTACACGGGGTCAGC-3′ Reverse 5′-CCTGCCAGCCTCGTTGTTGT-3′	This study
Gapdh	Rattus norvegicus	Forward 5′-TCTTGTGCAGTGCCAGCCTC-3′ Reverse 5′-CAAGAGAAGGCAGCCCTGGT-3′	This study
CypA	Rattus norvegicus	Forward 5′-TATCTGCACTGCCAAGACTGAGTG-3′ Reverse 5′-CTTCTTGCTGGTCTTGCCATTCC-3′	[58]
axin2	Danio rerio	Forward 5′-CAGAAGTGGCCTTGGGGCAT-3′ Reverse 5′-TGGCAGCTGGAGGAGACTGT-3′	This study
wnt4	Danio rerio	Forward 5′-GCCATCGACGAGTGCCAGTA-3′ Reverse 5′-ATGCAGCTTCCCTCGTACCTTG-3′	This study
col1a1a	Danio rerio	Forward 5′-TATTGG TGG TCA GCGTGGTA-3′ Reverse 5′-TCCTGG AGT ACC CTCACGAC-3′	[59]
col5a1	Danio rerio	Forward 5′-GATCCCAACCAGGGCTGCTC-3′ Reverse 5′-GGAGGTGAGTCTGGCCCCTT-3′	This study
Gapdh	Danio rerio	Forward 5′-CAGGCATAATGGTTAAAGTTGGTA-3′ Reverse 5′-CATGTAATCAAGGTCAATGAATGG-3′	[60]
ef1α1a	Danio rerio	Forward 5′-GAGACGCGGCCATTGTGGAA-3′ Reverse 5′-ACCGTCTGACGCATGTCACG-3′	This study
miR-34a-5p	Homo sapiens	5′-UGGCAGUGUCUUAGCUGGUUGU-3′	Qiagen:MS00003318
miR-21-5p	Homo sapiens	5′-UAGCUUAUCAGACUGAUGUUGA-3′	Qiagen:MS00009079
miR-181a-5p	Homo sapiens	5′-AACAUUCAACGCUGUCGGUGAGU-3′	Qiagen:MS00008827
miR-208a-3p	Homo sapiens	5′-AUAAGACGAGCAAAAAGCUUGU-3′	Qiagen:MS00003794
miR-133a-3p	Homo sapiens	5′-UUUGGUCCCCUUCAACCAGCUG-3′	Qiagen:MS00031423
miR-19b-3p	Homo sapiens	5′-UGUGCAAAUCCAUGCAAAACUGA-3′	Qiagen:MS00031584
RNU6-2_11	Homo sapiens	-	Qiagen:MS00033740
SNORD61_11	Homo sapiens	-	Qiagen:MS00033705

## Data Availability

The datasets used and/or analyzed during the current study are available from the corresponding author on reasonable request.

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
