# Peer review of "Tankyrase Inhibition Attenuates Cardiac Dilatation and Dysfunction in Ischemic Heart Failure"

_ijms, 2022, doi:10.3390/ijms231710059_

Round 1

Reviewer 1 Report

The manuscript by Wang et al examines the role of TNKS in pathogenesis of IHF by regulating WNt/beta catenin signalling pathway by modulating miRNAs and thereby conclude that TNK inhibition can represent a therapeutic potential for prevention of IHF.

The manuscript is well written and I have only minor concerns.

Fig 1A: There is huge variation within samples from IHF group in TNKS1. Can authors comment on that. 

Fig 1B: There is huge variation within samples from control group in AXIN1. Can authors comment on that.

Fig 1D: Sample variation has been seen in miR34a and miR21 where only one number is significantly elevated. Or else, without that one number, there will be no significantly difference. Is that an outlier? Can authors verify these results or simply repeat to see if this is how it looks.

TNKS2 is elevated in human samples and TNKS1 is elevated in rat samples. Can authors explain this discrepancy in results?

Author Response

Dear Reviewer,

We are grateful to the Reviewer for your valuable comments and suggestions on our manuscript. Below, please find our response to each of the comments.

  1. Fig 1A: There is huge variation within samples from IHF group in TNKS1. Can authors comment on that. 

We have quantitatively analyzed the expression of TNKS1 using GAPDH as an internal control. We found no huge variation in the relative expression of TNKS1 within samples from IHF group. The variation observed in TNKS1 band within samples from IHF group could be due to variations in pipetting and sample concentration, as the similar variation also appeared in GAPDH band.

  1. Fig 1B: There is huge variation within samples from control group in AXIN1. Can authors comment on that.

As pointed by the Reviewer, there were two samples showed slightly higher level of AXIN1 than other samples in control group in repeated western-blotting. Control samples were derived from the LVs of organ donors without cardiac disease, whose hearts could not be used as whole organ grafts due to tissue type or size mismatch. It may be that physical condition, medication, and aging affect the expression of AXIN1 in LVs.

  1. Fig 1D: Sample variation has been seen in miR34a and miR21 where only one number is significantly elevated. Or else, without that one number, there will be no significantly difference. Is that an outlier? Can authors verify these results or simply repeat to see if this is how it looks.

As pointed by the Reviewer, we observed considerable high levels of miR-34a-5p and miR-21-5p in one patient´s LV sample in repeated RT-qPCR. We performed outlier test with GraphPad Prism. This sample was not identified as an outlier. This patient had similar medical and medication history to other patients. Thus, we do not have the rationale for excluding this sample.

  1. TNKS2 is elevated in human samples and TNKS1 is elevated in rat samples. Can authors explain this discrepancy in results?

As suggested by the reviewer, a sentence In addition, the observation of elevated TNKS1 in MI rats and TNKS2 in patients’ LVs may indicate a major functional role of TNKS1 in rat heart and TNKS2 in human heart. has been included in the Discussion at page 16, line 384-386.

We hope that the revisions in the manuscript and accompanying responses could be suitable for publication in IJMS. We are willing to make further modifications to improve the manuscript if needed.

Sincerely,

Hong Wang, PhD

Reviewer 2 Report

The authors of the manuscript provide a well designed and also very well presented work with a large variety of underlying experimental methods to prove their hypotheses. 

There are only a few minor points which might be addressed:

1.) The part of the statistical description is too short in my opinion. In some experiments the number of trials and/or subjects is too small for descriptive analysis with mean+SD, please give alternative measures such as medians or IQR in these cases.

2.) Maybe add a small section to potential future trials in whole-heart models of larger animals to gain more insight into pharmacological effects of Tankyrase inhibition.

3.) Could you please report whether you found any side effects of inhibition in the experiments? As you suggest potential future use in humans, this might be of interest to the readers.

Author Response

Dear Reviewer,

We are grateful to the Reviewer for your valuable comments and suggestions on our manuscript. Below, please find our response to each of the comments.

1.) The part of the statistical description is too short in my opinion. In some experiments the number of trials and/or subjects is too small for descriptive analysis with mean+SD, please give alternative measures such as medians or IQR in these cases.

As suggested by the reviewer, more detailed description for statistical analyses have been added to Method and Materials, p21, line 642-645. In addition, the medians of the group with the sample number smaller than five have been included in Results.

2.) Maybe add a small section to potential future trials in whole-heart models of larger animals to gain more insight into pharmacological effects of Tankyrase inhibition.

We have included a sentence Such potential could be confirmed in an in vivo mammalian model to gain more translational insight into pharmacological effects of TNKS inhibition in the future.” in Discussion, p18, line 469-471.

3.) Could you please report whether you found any side effects of inhibition in the experiments? As you suggest potential future use in humans, this might be of interest to the readers.

As suggested by the Reviewer, a sentence Administering 10 μM of XAV939 to zebrafish had no effect on viability, swimming activity, or obvious morphological phenotype. has been added to Method and Material, p19, line 531-532.

We hope that the revisions in the manuscript and accompanying responses could be suitable for publication in IJMS. We are willing to make further modifications to improve the manuscript if needed.

Sincerely,

Hong Wang, PhD

Reviewer 3 Report

The paper is well-written and make a significant contribution in understanding roles of TNKS in MI. I just have very minor comments. Thanks

Please spell out acronym s at first mentions. E.g., p.2, ln.87: "LV".

p.3 & 7 & 13: please remove the blank space.

Author Response

Dear Reviewer,

We are grateful to the Reviewer for your valuable suggestions on our manuscript. Below, please find our response to each of the comments.

  1. Please spell out acronym s at first mentions. E.g., p.2, ln.87: "LV".

The acronyms at first mentions have been spell out.

  1. 3 & 7 & 13: please remove the blank space.

The blank spaces have been removed.

Sincerely,

Hong Wang PhD